## Research Article

psychological distress; gender; sustainable development goals; emerging adults; COVID19; substance use; adverse childhood experiences

**Corresponding author:**
Varalakshmi Chandra Sekaran;
Email: varalakshmi.cs@manipal.edu

# Gender-based determinants of psychological distress among emerging adult students in coastal Karnataka: a cross-sectional study

Ajith K. Remesan[1], Varalakshmi Chandra Sekaran[1] ,

Teddy Andrews Jaihind Jothikaran[2], Anish V. Cherian[3] and Lena Ashok[2]

[1]Department of Global Public Health Policy and Governance, Prasanna School of Public Health, Manipal Academy of Higher Education, Manipal, India; [2]Department of Social and Health Innovation, Prasanna School of Public Health, Manipal Academy of Higher Education, Manipal, India and [3]Department of Psychiatric Social Work and NIMHANS - Suicide Prevention, Research, Implementation, Training and Engagement (N-SPRITE) Centre, National Institute of Mental Health and Neurosciences (NIMHANS), Bengaluru, Karnataka, India

## Abstract

*Background*: Psychological distress is one of the major public health challenges during the emerging adulthood period, which is a developmental stage characterized by major life transformations and instability. Emerging adults are more vulnerable to psychological distress as they frequently deal with different pressures about relationships, work, education and identity exploration. The rising prevalence of psychological distress can impair physical health and wellbeing along with the initiation of harmful behaviors such as substance use. *Aim:* The study aims to explore the prevalence and associated factors of psychological distress among emerging adult students, comparing female and male participants. *Method:* Data were collected from 957 emerging adults in 12 institutions under Mangalore University in the Udupi district of Karnataka, India, pursuing bachelor's degrees. Along with the sociodemographic pro forma, the Kessler Psychological Distress Scale (K10) was used to measure psychological distress, the World Health Organization's Alcohol, Smoking and Substance Involvement Screening tool (WHO ASSIST version 3.0) was used to measure substance use, and the Adverse Childhood Experiences Scale was used to identify adverse childhood experiences. Univariate and multivariate regression analysis were employed to understand the determinants of psychological distress among participants. *Results:* Female participants reported a higher risk of experiencing psychological distress compared to male participants. Overall, 27.06% of participants experienced psychological distress, with 11.8% likely to have mild, 9.71% moderate and 5.53% severe mental disorders. While adverse childhood experiences were reported to be a major factor associated with psychological distress among both the groups (AOR 6.218, 95% CI (3.546, 10.901), $p < 0.001$, for female), (AOR 1.965, 95% CI (1.073, 3.601), $p = 0.029$, for male) substance use pattern during the COVID19 pandemic was also predicted higher psychological distress among male participants. *Conclusion:* In the study setting, psychological distress was prevalent among emerging adults, with a higher incidence among females. Adverse childhood events and substance use further elevated vulnerability. These findings highlight the critical need for culturally relevant and gender-sensitive mental health interventions.

## Impact statement

The study fills a significant gap in understanding the prevalence of psychological distress among female and male emerging adult students in coastal Karnataka. The study also assessed the various factors associated with psychological distress among both groups, including sociocultural and economic factors. The study setting has its cultural uniqueness, including rural–urban communities, coastal areas and different sociocultural backgrounds.

The study highlights that the prevalence of psychological distress was 32.58% among female participants and was 20.19% among male participants. These results highlight the need for gender-sensitive interventions in association with mental health issues, and they will contribute to promoting overall well-being, as mentioned in Sustainable Development Goal (SDG) 3.4. Female participants studied in the second or third year of college had higher psychological distress, which demands the need for educational institutions to have mental health programs. Adverse childhood experiences and substance use among the participants also acted as a risk factor for developing psychological distress, which is a global issue and needs to be addressed effectively. The research gap in the association between adverse childhood experiences, psychological distress and addiction is an area that requires further exploration.

## Introduction

Psychological distress, a broad term that includes symptoms of general stress, depression, posttraumatic stress disorder and anxiety (Zhu et al., 2022), has widely been recognized as a significant public health concern among emerging adults in the recent past (Pilar Matud et al., 2020; Hockey et al., 2022). Emerging adulthood, spanning 18 to 25 years, is a distinct phase in each individual's life marked by self-exploration, identity exploration, instability, optimism about the future and the shift from adolescence to adulthood (Arnett, 2000).

Whereas gender in the context of emerging adulthood refers to the socially developed roles, attributes and behaviors that society considers appropriate, generally categorized as male or female, rather than their biological sex. These gender roles and identities are shaped by cultural norms, expectations and socialization processes. These characteristics can increase their vulnerability to psychological distress (Matud et al., 2023; Conde et al., 2024).

Supporting these factors, global studies found a prevalence of 57% for psychological distress among youth, with 24.6% at mild psychological distress, 18% with moderate psychological distress and 14.4% had severe psychological distress (Anyanwu, 2023). While another study reported the prevalence of depression, stress and anxiety was 26.2% (10% mild, 7.4% moderate, 3.9% severe and 4.9% very severe), 29.7% (12.1% mild, 9.7% moderate, 5.8% severe and 2.1% very severe) and 39.9% (10.7% mild, 16% moderate, 4.2% severe and 9% very severe), respectively (Ismail et al., 2021).

In India, a study conducted across tier-1 cities in the country (Delhi, Hyderabad, Pune, Chennai, Ahmedabad, Bangalore, Kolkata, Mumbai) found a higher prevalence of psychological distress, with 63.8% reporting moderate distress and 6.8% of emerging adults reporting high distress (Suresh and Dar, 2025). Another study conducted in southern India reported 34.8% prevalence of psychological distress (TS et al., 2017). Mild and moderate psychological distress have important practical effects as they are linked to a greater likelihood of disrupting day-to-day functioning and may lower quality of life. To prevent those symptoms from developing into severe psychological distress, early intervention is needed (McLachlan and Gale, 2018).

Many research highlights various sociocultural aspects, such as gender, substance use, type of family and level of education, that were linked to the higher prevalence of psychological distress among emerging adults (Matud et al., 2023; Mirzaei-Alavijeh et al., 2025). Moreover, adverse childhood experiences (ACEs) and the use of substances were identified as significant risk factors for psychological distress among emerging adults (Tzouvara et al., 2023; Remesan et al., 2025a).

Additionally, studies also report various gender related pathways to psychological distress. Low decision-making opportunities among women appear as a prominent gendered stressor (Bilodeau et al., 2020). Women also report poorer self-esteem than men, which heightens their vulnerability to depression and anxiety, with self-esteem, social support and sleeplessness functioning as significant factors in female pathways (Beauregard et al., 2018). Men, on the other hand, are more susceptible to the effects of emotional regulation. ACEs have an indirect impact on anxiety and depression in both genders by lowering social support and self-esteem (Zhang et al., 2025).

In addition, the United Nations Sustainable Development Goals (SDGs) outline global priorities that align with the growing concern of psychological distress (Votruba et al., 2016). With the specific goal of reducing mental health issues and early mortality from non-communicable diseases by one-third by 2030, mainly focusing on women, SDG 3.4 places a strong emphasis on enhancing mental health and well-being (Bennett et al., 2020). Since social determinants such as inequality, access to education, work and poverty highly impact the mental health of women, addressing psychological distress is also linked to other SDGs (Kirkbride et al., 2024). Many studies worldwide indicate that women are more susceptible to psychological distress compared to men (Butterworth et al., 2020; Dessai et al., 2024). However, scientific enquiries on the psychological distress among the emerging adults through a gender lens in low and middle-income countries, including India has been inadequate.

In accordance with the stress diathesis model, psychological distress is caused by a combination of environmental stressors as well as individual vulnerabilities (Colodro-Conde et al., 2017). The risk of psychological distress among emerging adults in India can be elevated due to an array of external stressors like academic pressure, familial expectations and socioeconomic problems, as well as psychological histories such as ACEs or personality traits (Giano et al., 2021; Matud et al., 2023). This model emphasizes that stressors and vulnerabilities work together in affecting mental health outcomes rather than either one alone causing distress. This interaction underscores the significance of addressing sociocultural issues and provides a valuable framework for examining how various factors contribute to psychological distress among emerging adults (Taylor and Treur, 2023).

A notable scarcity of research examining psychological distress among emerging adults in India, particularly in the coastal Karnataka's rural–urban groups within various sociocultural contexts. These include gender roles, parental control fueled by cultural norms and societal expectations, which often heighten distress (Bhat et al., 2018). Additionally, the perspective about mental health concerns in this region is also influenced by cultural beliefs, such as supernatural forces. These beliefs often delay treatment and raise stigma, resulting in social isolation and discrimination (Hegde and Karkal, 2022). Furthermore, research exploring the factors such as ACEs and substance use associated with psychological distress are limited. Specifically in the Indian context, comparing these factors between genders leave a significant gap in understanding the underlying risk and protective factors.

Against this background, the current study aimed at understanding the prevalence and determinants of psychological distress among female and male emerging adult college students in Udupi district of Karnataka state in India. The study hypothesized that the prevalence of psychological distress would be higher among female participants in comparison with male participants. In keeping with existing literature, the study also hypothesized that adverse childhood experiences would emerge as a major determinant of psychological distress across both genders.

## Methods

### Participants and procedure

This cross-sectional study was conducted among undergraduate colleges affiliated with Mangalore University in the Udupi district, Karnataka, India. All procedures involving human participants were approved by the Institutional Ethics Committee of Kasturba Medical College and Kasturba Hospital, Manipal, Karnataka, India (IEC1:378/2022). The Registrar of Mangalore University approved the research, allowing it to be carried out in the associated colleges located in the Udupi district. Permissions were secured from all participating colleges involved in data collection. Data collection

occurred between June and October 2023, targeting emerging adults enrolled in undergraduate courses.

A stratified sampling approach was adopted by classifying colleges into three categories: government (fully funded and managed by the government), aided (private educational institution that receives financial support, such as grants, from the government for operational costs) and private (educational institutions owned and operated by private organizations or individuals). Using the university's website, a comprehensive list of 12 government colleges, 10 aided colleges and 12 private colleges in the district was prepared. From this pool, five government colleges, five aided colleges and two private colleges were selected through simple random sampling, with consideration given to the variety of academic streams and proportional student representation.

For determining the sample size, an initial calculation yielded 385 participants. However, the finite population correction was applied since the target population was estimated at around 18,000 students. Additionally, the adjusted calculation incorporated a design effect of two and an anticipated 20% non-response rate, leading to a revised sample size of 944. With approval from the selected institutions, data were gathered across various academic streams, including science, arts and commerce, treating each classroom as a sampling cluster. These classrooms were randomly selected based on different academic streams and representation of first year, second year and third year. Every student who was present in the selected classes at the time of data collection participated, and there were no missing values for any of the variables analyzed. Written informed consent was obtained from all the participants prior to data collection. Equal representation of female and male participants was ensured during data collection. On average, each classroom comprised 60–70 emerging adults. Responses were obtained from 957 students, aligning with the cluster sampling strategy employed (Figure 1).

### Measures

#### Sociodemographic pro-forma

The data were collected using a sociodemographic pro forma designed to capture the respondents' key socio-economic and cultural characteristics. The variables included age (categorized as 18–20 years and 21–25 years), year of study (first, second or third year) and type of educational institution (aided, government or private). The stream of study was classified into arts, science and commerce. Information was also gathered on living arrangements (living with parents or others), types of family (nuclear or joint/extended) and family's economic status (above or below the poverty line). Additional details included religion, number of siblings, birth order and primary caregiver during the first five years of life. The pro forma further captured the current marital status of parents and the participants' substance use patterns during the COVID-19 pandemic, such as whether use was initiated, increased or decreased, along with substance use among family members and peer groups.

#### The Kessler Psychological Distress Scale (K10)

The scale was used to assess the psychological distress among the participants (Kessler, 2002). The 10-question survey on psychological distress enquires about symptoms of depression and anxiety faced by participants in the last one month, culminating in an aggregate distress score. The cumulative psychological distress score of the Kessler Scale is calculated by summing the values assigned to each participant's response. Scores range from 10 to 50. According to this scale, a score below 20 suggests that a participant is likely in a stable psychological state. Scores from 20 to 24 are likely to have a mild disorder, 25 to 29 are likely to have a moderate disorder and 30 to 50 are likely to have a severe disorder.

#### The World Health Organization Alcohol, Smoking and Substance Involvement Screening Test version three (WHO ASSIST)

According to WHO, "the WHO ASSIST tool focused on substance involvement among emerging adults. The substances included alcohol, tobacco, cannabis, amphetamine-type stimulants, cocaine, inhalants, hallucinogens, sedatives or sleeping pills, opioids and others. These domains helped us to understand the substance use prevalence, and the total score of each domain guided our understanding of the emerging adults at moderate risk and high risk of health and other issues, such as social, legal, financial and relationship issues, due to the participants' pattern of substance use. Higher scores for each substance indicated a higher risk. For alcohol, scores from 0 to 10 represented a low-risk group, 11–26 represented a

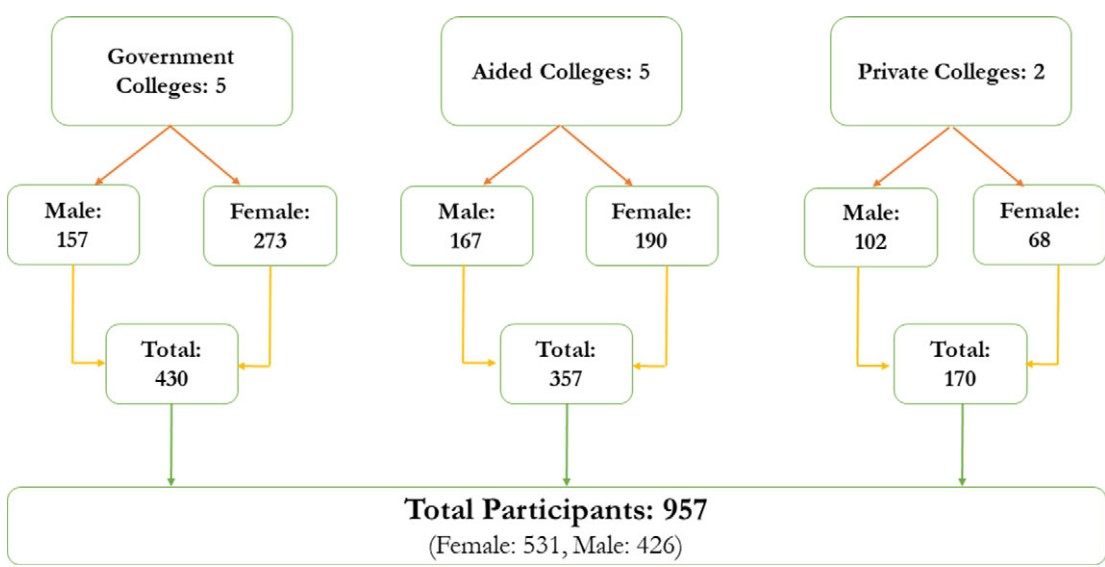

**Figure 1.** Sample distribution (Remesan et al., 2025b).

moderate-risk group and scores of 27 and more represented a high-risk group. For all the other substances, scores from 0 to 3 were considered low risk, 4–26 were considered moderate risk and 27 and above were considered high-risk groups" (WHO, 2013).

### Adverse childhood experiences (ACEs)

"The ACE questionnaire had ten questions, each answered by selecting "yes or no". The first three questions refer to emotional, physical and sexual abuse, respectively. Questions 4 and 5 assess emotional and physical neglect, and 6–10 refer to household dysfunctions of the participants" (Felitti et al., 1998). Surveys were administered in both English and Kannada languages, with the Kannada version undergoing reverse translation to ensure instrument reliability and consistency.

### Statistical analysis

Descriptive statistics were computed for all variables. Relationships between variables were examined using chi-square and Fisher's exact tests, with statistical significance as $p < 0.05$. Subsequently, univariate and multivariate regression analysis were employed using the Jamovi version 2.4.11 to understand the determinants of psychological distress. The variables that showed significance ($p < 0.2$) during the univariate analysis were adjusted in the multivariate analysis to determine the major factors associated with psychological distress (Chowdhury and Turin, 2020). Multicollinearity was assessed using variance inflation factors (VIF), and all values were below the standard threshold of five (Akinwande et al., 2015).

### Results

Among the 957 participants, 55.49% were female participants. Among the female participants, 89.1% belonged to 18–20 years of age and 54.2% were studying in the first year of college. The majority of them were studying in government colleges (51.4%) and 50.8% were studying the commerce stream. Also, 80.4% were living with both parents and 71.8% were from nuclear families. Additionally, 78.5% belonged to below the poverty line (INR 27000 per year per household). The majority of them reported that they have one or more siblings (90.6%) and most of them had their mother as homemakers (86.3%). In addition, 96% had their parents as primary caregivers, while 92.3% had their parents currently married.

Among the male participants, 81.7% belonged to the 18–20 years of age and 54.2% were studying in the second or third year of college. The majority of them were studying the commerce

stream (63.8%), and 81% were living with both parents. Additionally, 77.2% were from nuclear families, and 73.7% belonged to below the poverty line. Also, 88.7% had one or more siblings and 87.1% had mothers who were homemakers. In addition, 97.9% reported that they had parents as primary caregivers and 92% have their parents currently married. Among those who reported ACEs, 53.59% were female participants. Additionally, among those who reported substance use, 62.21% were male participants.

The study found an association between female participants' psychological distress and the socio-demographic variables. Specifically, age was associated with psychological distress with an effect size of $\varphi = 0.104$ and $p < 0.05$. Also, an association was found between the living arrangement of the female participants and psychological distress, showing an effect size $\varphi = 0.143$ and $p < 0.001$. Additionally, the type of female participants' educational institution showed an association with psychological distress, with an effect size $\varphi = 0.113$ and $p < 0.05$. In addition, an association was also found between the year in which the female participants were studying and psychological distress ($\varphi = 0.17$, $p < 0.001$).

Further, the religion of the female participants showed an association with psychological distress, with an effect size $\varphi = 0.1$ and $p < 0.05$, while the current marital status of parents also showed an association with psychological distress of female participants ($\varphi = 0.12$, $p < 0.05$). Moreover, substance use initiation among female participants during COVID-19 showed an association with psychological distress, with an effect size $\varphi = 0.1$ and $p < 0.05$. Peer substance use also reflected an association with psychological distress ($\varphi = 0.18$, $p < 0.001$). An association was found between ACEs of female participants with psychological distress, with a moderate effect size ($\varphi = 0.35$, $p < 0.001$).

Among male participants, age was associated with psychological distress, exhibiting an effect size $\varphi = 0.17$ and $p < 0.001$. Additionally, the year of study demonstrated an effect size $\varphi = 0.19$ and $p < 0.001$, showing an association with psychological distress. Moreover, an association was found between economic status and type of educational institute, with an effect size $\varphi = 0.10$ and $p < 0.05$ and Cramér's $V = 0.12$, $p < 0.05$, respectively. Furthermore, substance use was associated with psychological distress ($\varphi = 0.14$, $p < 0.05$), as were increased substance use during COVID-19 ($\varphi = 0.18$, $p < 0.05$) and decreased substance use during this period ($\varphi = 0.15$, $p < 0.05$). In addition, peer substance use ($\varphi = 0.15$, $p < 0.05$) and ACEs ($\varphi = 0.16$, $p < 0.001$) were also shown to be associated with psychological distress (Table 1).

The overall prevalence of psychological distress was 27.06%, with 11.8% likely to have a mild mental disorder, 9.71% likely to

**Table 1.** Socio-demographic details of the participants in association with psychological distress (K10) ($n = 957$)

| | | Psychological distress | | | | | | | |
| | | Female (531) | | | | Male (426) | | | |
| Sociodemographic variables | | At risk (%) | No risk (%) | $X^2$ (df) | p-value | At risk (%) | No risk (%) | $X^2$ (df) | p-value |
|---|---|---|---|---|---|---|---|---|---|
| Age | 18–20 years | 146 (30.87) | 327 (69.13) | 5.79 (1) | 0.016 | 59 (16.95) | 289 (83.05) | 12.34 (1) | <0.001 |
| | 21–25 years | 27 (46.55) | 31 (53.45) | | | 27 (34.62) | 51 (65.38) | | |
| Year of study | First year | 73 (25.35) | 215 (74.65) | 14.99 (1) | <0.001 | 23 (11.79) | 172 (88.21) | 15.72 (1) | <0.001 |
| | Second year/Third year | 100 (41.15) | 143 (58.85) | | | 63 (27.27) | 168 (72.73) | | |
| Type of educational institution | Aided | 63 (33.16) | 127 (66.84) | | 0.035 | 27 (16.17) | 140 (83.83) | | 0.048 |
| | Government | 97 (35.53) | 176 (64.47) | 6.72 (2) | | 30 (19.11) | 127 (80.89) | 6.09 (2) | |
| | Private | 13 (19.12) | 55 (80.88) | | | 29 (28.43) | 73 (71.57) | | |

*(Continued)*

**Table 1.** (*Continued*)

| | | Psychological distress | | | | | | | |
| --- | --- | --- | --- | --- | --- | --- | --- | --- | --- |
| | | Female (531) | | | | Male (426) | | | |
| Sociodemographic variables | | At risk (%) | No risk (%) | $X^2$ (df) | p-value | At risk (%) | No risk (%) | $X^2$ (df) | p-value |
| Stream of course | Arts | 72 (34.12) | 139 (65.88) | | 0.54 | 30 (21.13) | 112 (78.87) | | 0.68 |
| | Commerce | 88 (32.59) | 182 (67.41) | 1.21 (2) | | 55 (20.22) | 217 (79.78) | 1.12 (2) | |
| | Science | 13 (26) | 37 (74) | | | 1 (8.33) | 11 (91.67) | | |
| Living arrangement | Living with parents | 125 (29.27) | 302 (70.73) | 10.85 (1) | <0.001 | 71 (20.58) | 274 (79.42) | 0.17 (1) | 0.67 |
| | Others | 48 (46.15) | 56 (53.85) | | | 15 (18.52) | 66 (81.48) | | |
| Type of family | Nuclear | 125 (32.81) | 256 (67.19) | 0.03 (1) | 0.85 | 70 (21.28) | 259 (78.72) | 1.06 (1) | 0.30 |
| | Joint/extended | 48 (32) | 102 (68) | | | 16 (16.49) | 81 (83.51) | | |
| Economic status | Above poverty line | 36 (31.58) | 78 (68.42) | 0.07 (1) | 0.79 | 30 (26.79) | 82 (73.21) | 4.11 (1) | 0.043 |
| | Below poverty line | 137 (32.85) | 280 (67.15) | | | 56 (17.83) | 258 (82.17) | | |
| Believe in religion | Yes | 164 (31.97) | 349 (68.03) | 2.57 (1) | 0.10 | 81 (19.80) | 328 (80.20) | 0.94 (1) | 0.33 |
| | No | 9 (50) | 9 (50) | | | 5 (25.00) | 15 (75.00) | | |
| Religion | Hindu | 153 (31.22) | 337 (68.78) | 5.31 (1) | 0.021 | 73 (19.31) | 305 (80.69) | 1.6 (1) | 0.20 |
| | Others | 20 (48.78) | 21 (51.22) | | | 13 (27.08) | 35 (72.92) | | |
| Number of siblings | No siblings | 22 (44) | 28 (56) | 3.28 (1) | 0.07 | 9 (13.43) | 58 (86.57) | 2.25 (1) | 0.13 |
| | One or more siblings | 151 (31.39) | 330 (68.61) | | | 77 (21.45) | 282 (78.55) | | |
| Birth order | First born | 88 (33.21) | 177 (66.79) | 0.10 (1) | 0.75 | 42 (17.28) | 201 (82.72) | 2.96 (1) | 0.08 |
| | Second born/other | 85 (31.95) | 181 (68.05) | | | 44 (24.04) | 139 (75.96) | | |
| Mothers occupation | Working | 27 (36.99) | 46 (63.01) | 0.75 (1) | 0.38 | 14 (25.45) | 41 (74.55) | 1.087 (1) | 0.29 |
| | Home maker | 146 (31.88) | 312 (68.12) | | | 72 (19.41) | 299 (80.59) | | |
| Primary caregivers of participants (0–5 years) | Parents | 165 (32.35) | 345 (67.65) | 0.30 (1) | 0.58 | 84 (20.14) | 333 (79.86) | 0.02 (1) | 1 |
| | Others | 8 (38.1) | 13 (61.9) | | | 2 (22.22) | 7 (77.78) | | |
| Marital status of parents | Married | 152 (31.02) | 338 (68.98) | 7.03 (1) | 0.008 | 81 (20.66) | 311 (79.34) | 0.69 (1) | 0.40 |
| | Divorced/Separated/Widowed | 21 (51.22) | 20 (48.78) | | | 5 (14.71) | 29 (85.29) | | |
| Substance Use | Yes | 28 (43.08) | 37 (56.92) | 3.76 (1) | 0.05 | 32 (29.91) | 75 (70.09) | 8.38 (1) | 0.004 |
| | No | 145 (31.12) | 321 (68.88) | | | 54 (16.93) | 265 (83.07) | | |
| Substance use initiation during COVID–19 | Yes | 5 (71.43) | 2 (28.57) | 4.87 (1) | 0.040 | 7 (26.92) | 19 (73.08) | 0.78 (1) | 0.37 |
| | No | 168 (32.06) | 356 (67.94) | | | 79 (19.75) | 321 (80.25) | | |
| Substance use increase during COVID–19 | Yes | 1 (50) | 1 (50) | 0.28 (1) | 0.54 | 7 (63.64) | 4 (36.36) | 13.23 (1) | 0.002 |
| | No | 172 (32.51) | 357 (67.49) | | | 79 (19.04) | 336 (80.96) | | |
| Substance use decrease during COVID–19 | Yes | 0 (0) | 2 (100) | 0.97 (1) | 1 | 5 (62.50) | 3 (37.50) | 9.06 (1) | 0.01 |
| | No | 356 (67.3) | 173 (32.7) | | | 81 (19.38) | 337 (80.62) | | |
| Substance use by siblings | Yes | 7 (43.75) | 9 (56.25) | 0.94 (1) | 0.33 | 9 (34.62) | 17 (65.38) | 3.58 (1) | 0.05 |
| | No | 166 (32.23) | 349 (67.77) | | | 77 (19.25) | 323 (80.75) | | |
| Substance use by other family members | Yes | 28 (41.79) | 39 (58.21) | 2.96 (1) | 0.08 | 14 (22.22) | 49 (77.78) | 0.19 (1) | 0.66 |
| | No | 145 (31.25) | 319 (68.75) | | | 72 (19.83) | 291 (80.17) | | |
| Peer substance use | Yes | 38 (54.29) | 32 (45.71) | 17.29 (1) | <0.001 | 50 (27.32) | 133 (72.68) | 10.14 (1) | 0.001 |
| | No | 135 (29.28) | 326 (70.72) | | | 36 (14.81) | 207 (85.19) | | |
| Adverse childhood experiences | Yes | 65 (67.01) | 32 (32.99) | 64.05 (1) | <0.001 | 28 (33.33) | 56 (66.67) | | <0.001 |
| | No | 108 (24.88) | 326 (75.12) | | | 58 (16.96) | 284 (83.04) | 11.22 (1) | |

*Notes:* At risk: participants having Kessler Psychological Distress Scale (K10) score from 20 and above.
No risk: participants having Kessler Psychological Distress Scale (K10) score below 20.

**Table 2.** Psychological distress (K10) among emerging adults—*n* = 957 (female = 531, male = 426)

| Psychological distress | Frequency | |
|---|---|---|
| | Female (%) | Male (%) |
| Balanced | 358 (51.29) | 340 (48.71) |
| Mild | 68 (60.18) | 45 (39.82) |
| Moderate | 69 (74.19) | 24 (25.81) |
| Severe | 36 (67.92) | 17 (32.08) |

have a moderate mental disorder, and 5.53% likely to have a severe mental disorder. Psychological distress was more common among female participants (32.58%) than male participants (20.19%). Among participants with a likelihood of mild mental disorder, 60.18% were female. Among those with a likelihood of moderate mental disorder, 74.19% were female. Similarly, the likelihood of having a severe mental disorder was higher among females, who accounted for 67.92% of this group (Table 2).

Backward logistic regression was performed to find the predictors of the outcome variable, psychological distress, among female and male participants. Univariate and multivariate analyses for determinants of psychological distress were conducted. For female participants, factors such as age, year in which they are studying, type of their educational institution, their living arrangement, whether they believe in religion, their religion, number of siblings, current marital status of parents, substance use, substance use initiation during COVID-19, substance use by family members, peer substance use and ACEs were adjusted in the multivariate analysis after being shown association in the univariate analysis. Among these, the year in which they are studying, the type of their educational institution, peer substance uses and ACEs showed association in multivariate analysis and crude odds ratio (COR), adjusted odds ratio (AOR) and confidence interval (CI) were reported (Table 3).

With reference to female participants studying in the first year of college, those who studied in the second year or third year of college had higher odds of psychological distress (AOR 1.817, 95% CI (1.173, 2.815), *p* = 0.008). Concerning the type of educational institution the female participants studied, with reference to those who studied in private colleges, those who studied in government colleges had higher odds of psychological distress (AOR 3.25, 95% CI (1.486, 7.106), *p* = 0.003). Peer substance use was associated

**Table 3.** Univariate and multivariate analysis for associations of psychological distress (K10) among female participants (*n* = 531)

| Socio-demographic variables | | At risk (%) | No risk (%) | *n* (100%) | COR (95% CI) | *p*-value | AOR (95% CI) | *p*-value |
|---|---|---|---|---|---|---|---|---|
| | | | | | Psychological distress | | | |
| Age | 18–20 years | 146 (30.87) | 327 (69.13) | 473 | Ref | | Ref | |
| | 21–25 years | 27 (46.55) | 31 (53.45) | 58 | 1.951 (1.124,3.386) | 0.018 | 0.807 (0.388,1.679) | 0.566 |
| Year of study | First year | 73 (25.35) | 215 (74.65) | 288 | Ref | | Ref | |
| | Second year/third year | 100 (41.15) | 143 (58.85) | 243 | 2.06 (1.425,2.977) | < 0.001 | 1.817 (1.173,2.815) | 0.008 |
| Type of educational institution | Aided | 63 (31.66) | 127 (68.34) | 190 | 2.099 (1.068,4.125) | 0.032 | 2.077 (0.939,4.594) | 0.071 |
| | Government | 97 (35.53) | 176 (64.47) | 273 | 2.332 (1.213,4.481) | 0.011 | 3.25 (1.486,7.106) | 0.003 |
| | Private | 13 (19.18) | 55 (80.82) | 68 | Ref | | Ref | |
| Living arrangement | Living with parents | 125 (29.27) | 302 (70.73) | 427 | Ref | | Ref | |
| | Others | 48 (46.15) | 56 (53.85) | 104 | 2.071 (1.336,3.21) | 0.001 | 1.814 (1.001,3.288) | 0.05 |
| Believe in religion | Yes | 164 (31.95) | 349 (68.05) | 513 | Ref | | Ref | |
| | No | 9 (50) | 9 (50) | 18 | 2.128 (0.829,5.461) | 0.116 | 1.766 (0.629,4.955) | 0.28 |
| Religion | Hindu | 153 (31.22) | 337 (68.78) | 490 | Ref | | Ref | |
| | Others | 20 (48.78) | 21 (51.22) | 41 | 2.098 (1.104,3.984) | 0.024 | 1.872 (0.876,1.003) | 0.106 |
| Number of siblings | No siblings | 22 (44) | 28 (56) | 50 | Ref | | Ref | |
| | One or more siblings | 151 (31.39) | 330 (68.61) | 481 | 0.582 (0.323,1.051) | 0.073 | 0.603 (0.309,1.178) | 0.139 |
| Marital status of parents | Married | 152 (31.02) | 338 (68.98) | 490 | Ref | | Ref | |
| | Divorced/separated/widowed | 21 (51.22) | 20 (48.78) | 41 | 2.335 (1.229,4.435) | 0.010 | 0.966 (0.394,2.37) | 0.941 |
| Substance use | Yes | 28 (43.08) | 37 (56.92) | 65 | 1.675 (0.987,2.842) | 0.056 | 0.895 (0.425,1.883) | 0.77 |
| | No | 145 (30.46) | 321 (69.54) | 476 | Ref | | Ref | |
| Substance use initiation during COVID–19 | Yes | 5 (71.43) | 2 (28.57) | 7 | 5.298 (1.017,27.586) | 0.048 | 2.448 (0.312,19.184) | 0.394 |
| | No | 168 (32.06) | 356 (67.94) | 524 | Ref | | Ref | |
| Substance use by family members | Yes | 28 (41.79) | 39 (58.2) | 67 | 1.579 (0.936,2.667) | 0.087 | 0.576 (0.292,1.139) | 0.113 |
| | No | 145 (31.25) | 319 (68.75) | 464 | Ref | | Ref | |

(*Continued*)

**Table 3.** (*Continued*)

| Socio-demographic variables | | At risk (%) | No risk (%) | *n* (100%) | COR (95% CI) | *p*-value | AOR (95% CI) | *p*-value |
|---|---|---|---|---|---|---|---|---|
| | | | | | | | | |
| Peer substance use | Yes | 38 (54.29) | 32 (45.71) | 70 | 2.868 (1.72,4.781) | < 0.001 | 2.391 (1.249,4.577) | 0.009 |
| | No | 135 (29.28) | 326 (70.72) | 461 | Ref | | Ref | |
| Adverse childhood experiences | Yes | 65 (67.01) | 32 (32.99) | 97 | 6.131 (3.81,9.868) | < 0.001 | 6.218 (3.546,10.901) | <0.001 |
| | No | 108 (24.88) | 326 (75.12) | 434 | Ref | | Ref | |

*Notes:* At risk: participants having a Kessler Psychological Distress Scale (K10) score of 20 and above.
No risk: participants having a Kessler Psychological Distress Scale (K10) score below 20.
Abbreviations: AOR: adjusted odds ratio, COR: crude odds ratio, CI: confidence interval.

with psychological distress among female participants. Participants with peers using substances had higher odds of psychological distress (AOR 2.391, 95% CI (1.249,4.577), *p* = 0.009) than non-substance-using peers. ACEs were associated with psychological distress as female participants who experienced ACEs had higher odds of psychological distress (AOR 6.218, 95% CI (3.546, 10.901), *p* < 0.001) with reference to those who never experienced ACEs.

For male participants, factors such as age, year in which they are studying, type of their educational institute, economic status of family, number of siblings, birth order, substance use, increased substance use during COVID-19, reduced use of substances during COVID-19, substance use by siblings, peer substance use and ACEs were adjusted in the multivariate analysis after being shown association in the univariate analysis. Among those, year in which they are studying, type of their educational institute, increased substance use during COVID-19, reduced use of substances during COVID-19 and ACEs showed association in multivariate analysis and crude odds ratio (COR), adjusted odds ratio (AOR) and confidence interval (CI) were reported (Table 4).

With reference to male participants studying in the first year of college, those who studied in the second year or third year of college had higher odds of psychological distress (AOR 1.953, 95% CI (1.05, 3.634), *p* = 0.035). Concerning the type of educational institution the male participants studied, with reference to those who studied in private colleges, those who studied in aided colleges had lower odds of psychological distress (AOR 0.398, 95% CI (0.203, 0.781), *p* = 0.007). Among male participants, both an increase and a decrease in substance use during the COVID-19 pandemic were associated with a risk of psychological distress compared to those who reported no change or had not initiated substance use prior to the pandemic (AOR 5.588, 95% CI (1.28, 24.394) *p* = 0.022), (AOR 5.595, 95% CI (1.066, 29.359) *p* = 0.042), respectively. ACEs were also associated with psychological distress among male participants. Male participants who experienced ACEs had higher odds of psychological distress (AOR 1.965, 95% CI (1.073, 3.601), *p* = < 0.029) with reference to those who never experienced ACEs.

## Discussion

Addressing the gap in data on prevalence and determinants of psychological distress among female and male emerging adult students in coastal Karnataka, the current study found that female participants reported the higher odds of experiencing psychological distress (32.58%) compared to male participants (20.19%), a pattern that is consistently observed in research both globally and within India (Maser et al., 2019; Panigrahi et al., 2021; Viertiö et al., 2021). Even though multiple studies confirm that women report higher psychological distress (Mommersteeg et al., 2024; Jamil, 2025), this study advances novelty by focusing on emerging adults.

Particularly in coastal Karnataka, gender roles and parental control based on cultural norms can be a major cause of increased psychological distress (Bhat et al., 2018). Women are expected to uphold family honor, have less autonomy in making decisions and experience stress, including relationship problems and academic challenges, which can also be the reason for heightened psychological distress (Antabe et al., 2025; Shergill and Rathore Hooja, 2025). In addition, studies also reported that females' ability to understand minute emotional changes than that of males, which may result in underreporting of distress among males (Shi et al., 2021). This highlights the need for gender-sensitive approaches in mental health assessment and intervention, which will also help in promoting mental health and well-being, as mentioned in SDG 3.4.

Overall, 27.06% of participants experienced psychological distress, with 9.71% moderate and 5.53% severe mental disorders. This prevalence is similar to a study from coastal Karnataka, India, which reported a rate of 28.5% (Bhat et al., 2018). However, it is notably lower than the 57% prevalence in a global study among 906 students by Anyanwu (2023). These findings suggest that while psychological distress rates may vary across different populations (Nochaiwong et al., 2021), the results of the current study are consistent with regional data from India.

Compared to participants who had not experienced ACEs, those who experienced ACEs were more at risk of psychological distress, indicating that ACEs function as a major factor associated with psychological distress between both groups. Different studies also report the impact of ACEs on poor mental health outcomes (Tzouvara et al., 2023). The study, by assessing ACEs as a factor associated with psychological distress, fills a critical gap in mental health research in the region, offering targeted insights for tailored interventions.

Moreover, female and male participants studying in their second or third year of college had a higher risk of psychological distress compared to their counterparts in first year of college. This can be associated with existing literature, which reports that academic pressure intensifies as students progress into the final years (Alam et al., 2025). Students face higher demands of coursework, specialization, thesis work, career decisions and future uncertainties, which can heighten distress. Additionally, as students get older, increased responsibilities and expectations may further contribute to psychological distress, making upper-year students more vulnerable than their first-year peers (Jaisoorya et al., 2017).

**Table 4.** Univariate and multivariate analysis for associations of psychological distress (K10) among male participants (*n* = 426)

| Socio-demographic variables | | At risk (%) | No risk (%) | *n* (100%) | COR (95% CI) | *p*-value | AOR (95% CI) | *p*-value |
|---|---|---|---|---|---|---|---|---|
| | | | | | **Psychological distress** | | | |
| Age | 18–20 years | 59 (16.95) | 289 (83.05) | 348 | Ref | | Ref | |
| | 21–25 years | 27 (34.62) | 51 (65.38) | 78 | 2.593 (1.505,4.468) | < 0.001 | 1.647 (0.853,3.18) | 0.137 |
| Year of study | First year | 23 (11.79) | 172 (88.21) | 195 | Ref | | Ref | |
| | Second/third year | 63 (27.27) | 168 (72.73) | 231 | 2.804 (1.663,4.73) | < 0.001 | 1.953 (1.05,3.634) | 0.035 |
| Type of educational institution | Aided | 27 (16.17) | 140 (83.83) | 167 | 0.485 (0.268,0.881) | 0.017 | 0.398 (0.203,0.781) | 0.007 |
| | Government | 30 (19.11) | 127 (80.89) | 157 | 0.595 (0.331,1.068) | 0.082 | 0.662 (0.34,1.287) | 0.224 |
| | Private | 29 (28.43) | 73 (71.57) | 102 | Ref | | Ref | |
| Economic status | Above poverty line | 30 (26.79) | 82 (73.21) | 112 | 1.686 (1.014,2.802) | 0.044 | 1.255 (0.682,2.309) | 0.465 |
| | Below poverty line | 56 (17.83) | 258 (82.17) | 314 | Ref | | Ref | |
| Number of siblings | No siblings | 9 (13.43) | 58 (86.57) | 67 | Ref | | Ref | |
| | One or more siblings | 77 (21.45) | 282 (78.55) | 359 | 1.76 (0.835,3.71) | 0.138 | 2.632 (0.945,5.907) | 0.066 |
| Birth order | First born | 42 (17.28) | 201 (82.72) | 243 | Ref | | Ref | |
| | Second born/other | 44 (24.04) | 139 (75.97) | 183 | 1.515 (0.942,2.436) | 0.086 | 1.205 (0.687,2.114) | 0.515 |
| Substance use | Yes | 32 (29.91) | 75 (70.09) | 107 | 2.094 (1.261,3.476) | 0.004 | 1.099 (0.552,2.191) | 0.788 |
| | No | 54 (16.93) | 265 (83.07) | 319 | Ref | | Ref | |
| Substance use increase during COVID–19 | Yes | 7 (63.64) | 4 (36.36) | 11 | 7.443 (2.127,26.048) | 0.002 | 5.588 (1.28,24.394) | 0.022 |
| | No | 79 (19.04) | 336 (80.96) | 415 | Ref | | Ref | |
| Substance use decrease during COVID–19 | Yes | 5 (62.5) | 3 (37.5) | 8 | 6.934 (1.624,29.613) | 0.009 | 5.595 (1.066,29.359) | 0.042 |
| | No | 81 (19.38) | 337 (80.62) | 418 | Ref | | Ref | |
| Substance use by siblings | Yes | 9 (34.62) | 17 (65.38) | 26 | 2.221 (0.954,5.172) | 0.064 | 1.547 (0.585,4.091) | 0.379 |
| | No | 77 (19.25) | 323 (80.75) | 400 | Ref | | Ref | |
| Peer substance use | Yes | 50 (27.32) | 133 (72.68) | 183 | 2.162 (1.337,3.495) | 0.002 | 1.249 (0.698,2.234) | 0.454 |
| | No | 36 (14.81) | 207 (85.19) | 243 | Ref | | Ref | |
| Adverse childhood experiences | Yes | 28 (33.33) | 56 (66.67) | 84 | 2.448 (1.435,4.178) | 0.001 | 1.965 (1.073,3.601) | 0.029 |
| | No | 58 (16.96) | 284 (83.04) | 342 | Ref | | Ref | |

*Notes:* At risk: participants having a Kessler Psychological Distress Scale (K10) score of 20 and above.
No risk: participants having a Kessler Psychological Distress Scale (K10) score below 20.
Abbreviations: AOR: adjusted odds ratio, COR: crude odds ratio, CI: confidence interval.

Additionally, with reference to female participants who studied in private colleges, those who studied in government colleges were having higher risk of psychological distress. This finding contrasts with the majority of the existing literature reporting higher psychological distress among students from private colleges (Alkhlaiwi et al., 2023; Smritikana Mitra Ghosh, 2016). This could be due to different contextual factors such as the academic environment and the societal background of the participants.

Whereas, with reference to male part,icipants who studied in private colleges, male participants who studied in aided colleges had lesser odds of psychological distress. The results suggest that male participant data aligns with existing literature (Shafiq et al., 2021), exploring social support or academic environments in government colleges in the study area may be valuable, as it contributes to increased distress in female students, countering the pattern reported in other regions.

Among female participants, having peers who used substances was associated with a higher likelihood of experiencing psychological distress, compared to those whose peers did not engage in substance use, but this observation is not reported for male participants. This may reflect differences in socialization and coping strategies and the importance of peer approval and support among females (Yoon et al., 2023). The results also suggest that prevention programs should consider these gender differences in addressing the psychological risks associated with substance-using peer groups (Pickering et al., 2020).

Among male participants, both an increase and a decrease in substance use during the COVID-19 pandemic were associated with a higher risk of psychological distress compared to those who reported no change or had not initiated substance use prior to the pandemic. Substance use among emerging adults was associated with higher levels of psychological distress globally and in India in the

literature (TS et al., 2017; Connery et al., 2020). During the pandemic, studies have reported substance use as a maladaptive coping mechanism among emerging adults, which will, in turn, affect their mental health (Romano et al., 2021; Remesan et al., 2023).

The reason for higher psychological distress among those who decreased substance use can be forced or unintentional due to the external constraints, such as lockdown, and loss of access to substances. This may have disrupted routines or coping strategies, resulting in increased distress. Sudden reduction in substance use, especially if not planned or supported, can cause psychological distress due to withdrawal or loss of a familiar coping tool (Layman et al., 2022; Korakkottil et al., 2025).

In both genders, the associations between sociodemographic variables and psychological distress primarily show small effect sizes, suggesting a moderate but an association with practical implications. Age, year of study, type of educational institution, substance use – including changes during COVID-19, and peer substance use all indicated minor effects, suggesting that these factors marginally but constantly contribute to psychological distress (Roberts et al., 2021; Layman et al., 2022). The moderate effect size of ACEs in females is associated, indicating a greater practical importance for trauma-informed interventions (Haahr-Pedersen et al., 2020). These results highlight the need for comprehensive, gender-sensitive approaches among emerging adults, suggesting that although each element alone has a small effect, their combined influence is significant for targeted mental health promotion and intervention.

The study's results are consistent with the stress-diathesis model, in which ACEs act as a primary diathesis and have a moderate to small effect on psychological distress in emerging adult female and male participants, respectively (Mosley-Johnson et al., 2021). These effects are heightened by academic stressors, particularly in the second or third year of study.

Sociodemographic characteristics, including living arrangements, economic condition and type of college, function as intermediate stressors that interact with these vulnerabilities, especially increasing the risk of psychological distress in female participants (Ayad et al., 2024). Peer substance use and substance use initiation during COVID-19 are examples of substance use-related variables that served as major environmental triggers that contributed to diathesis (Chacon et al., 2021). Gender differences demonstrated that female participants were more vulnerable to social and institutional challenges.

These results highlight the complex ways in which gender, ACEs and substance use interconnect to influence psychological distress in emerging adults. By revealing the adverse effects of these factors on psychological distress among emerging adult female and male participants, the study underscores the need for early, culturally sensitive and multidimensional mental health interventions. Addressing these issues is essential for promoting well-being, reducing inequalities and achieving sustainable development in line with the SDGs.

## Conclusion

The study contributed to understanding the necessity for gender-sensitive approaches in youth mental health interventions, which is still not widely recognized in the Indian context. SDG 3.4, which aims to promote mental health and wellbeing, is in line with these findings, which demand gender-responsive and culturally sensitive mental health policies. Gender-sensitive policies should be given importance in mental health initiatives at educational institutions. Normalizing help-seeking behaviors, reducing stigma and boosting

access to mental health assistance are vital. The effectiveness of interventions across various populations should be evaluated in future studies. Gender-sensitive programs are also necessary to close the gap between male's underreporting symptoms and female's disproportionate burden of psychological distress associated with patriarchal stresses and discrimination (Vigod and Rochon, 2020; Shi et al., 2021).

## Limitation

Even though the current research provides insightful information, it is critical to acknowledge its limits. The sample was restricted to emerging adults who attended undergraduate university programs, which limits the findings' generalizability and the cross-sectional methodology does not allow for temporality. The scope of future research should be expanded to include emerging adults from different societal sectors, such as those who work or pursue other careers. A further limitation is using self-reported data, where the individuals may underreport psychological distress because of recall biases or social desirability, which could compromise the accuracy of the findings. To overcome these issues, longitudinal research is recommended, which helps to establish a causal relationship between psychological distress and its influencing factors and allows for the assessment of changes over time. By recognizing these shortcomings and implementing the suggestions into practice, future studies can offer a more thorough understanding of psychological distress among emerging adults.

**Open peer review.** To view the open peer review materials for this article, please visit http://doi.org/10.1017/gmh.2026.10156.

**Data availability statement.** The authors confirm that the data supporting the findings of this study are available within the article.

**Acknowledgements.** The authors would like to thank the heads and faculties of the colleges that were part of the study for their valuable time and support in data collection.

**Author contribution.** Ajith K. Remesan: Conceptualization, Data curation, Formal analysis, Funding acquisition, Investigation, Methodology, Resources, Software, Supervision, Validation, Writing – original draft, Writing – review & editing. Varalakshmi Chandra Sekaran: Conceptualization, Data curation, Formal analysis, Methodology, Resources, Software, Supervision, Validation, Writing – original draft, Writing – review & editing. Teddy Andrews Jaihind Jothikaran: Conceptualization, Methodology, Resources, Software, Supervision, Validation, Writing – review & editing. Anish V Cherian: Conceptualization, Resources, Supervision, Validation, Writing – review & editing. Lena Ashok: Conceptualization, Data curation, Methodology, Resources, Software, Supervision, Validation, Writing – review & editing.

**Financial support.** I acknowledge the Ministry of Social Justice & Empowerment, Government of India, for providing the fellowship support that enabled me to undertake and complete this research. No. F. 82–44/2020 (SA-III) UGC NFSC JRF funded by Ministry of Social Justice & Empowerment, Government of India.

**Competing interests.** The authors declare none.

**Ethical standard.** All procedures involving human participants were approved by the Institutional Ethics Committee of Kasturba Medical College and Kasturba Hospital, Manipal, Karnataka, India (IEC1:378/2022). The Registrar of the Mangalore University approved the research, allowing it to be carried out in the associated colleges located in the Udupi district. Written informed consent was obtained from all the participants prior to the data collection.

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
