## [Reviewer Report]

Dear Authors,

Thank you for the opportunity to review your manuscript titled “Gender Differences in Prevalence and Predictors of Psychological Distress among Emerging Adults in Udupi District, Coastal Karnataka: A Cross-Sectional Study.”

At the outset, I would like to commend your team on undertaking a highly relevant and socially meaningful study. The manuscript addresses an important mental-health problem among emerging adults, uses widely accepted tools (K10, WHO-ASSIST, ACEs), and benefits from a large sample size and rigorous fieldwork. The focus on gender differences, ACEs, and substance use patterns during COVID-19 adds considerable value to the evidence base for Indian youth mental health. The linkage to SDGs and the contextualization to Udupi district also enrich the paper.

While the study has strong potential for publication, several revisions—mainly to improve scientific rigor, clarity, precision, and interpretation—will further strengthen the manuscript.

Below are my detailed comments and suggestions:

1. Clarify Study Objectives & Research Questions

The introduction mentions prevalence and predictors, but specific research questions or hypotheses are not clearly articulated.

Consider framing them more explicitly.

Clear research questions will strengthen methodological alignment.

2. Justify the Focus on Gender Comparison

You report gender differences, but the introduction should better theorize why these differences matter. Please consider:

• Expanding the literature on gender-based pathways to distress

• Clarifying if gender interactions were formally tested (interaction terms in regression)

• Explaining cultural/sociological reasons specific to coastal Karnataka

This will deepen theoretical coherence.

3. Improve Sampling Clarity and Flow

The sampling description is detailed but could be simplified and clarified. Suggested improvements:

• Provide a flow diagram of college selection → classroom selection → participant inclusion

• Justify why only 2 private colleges were included (possible sampling imbalance)

• Clarify how randomization was ensured in classroom selection

• Mention whether response bias was assessed since there were no missing values, which is uncommon in field surveys

4. Regression Modelling – Improve Scientific Rigor

The regression section requires more methodological clarity:

• Explain why stepwise backward logistic regression was chosen, as this method has statistical limitations (risk of overfitting, instability of estimates).

• Mention whether multicollinearity was tested (e.g., VIF).

• Clarify how confounders were selected for the final multivariate model.

• Some AOR confidence intervals appear incorrect or incomplete (e.g., ACEs for females shows “CI (3.546, 1.901)”—should be ascending order).

• Interpretation of AORs should be more nuanced, acknowledging potential suppression or confounding effects.

5. Recheck Statistical Interpretation

Some findings are described as “highly significant” but require clearer statistical language.

Suggestions:

• Report exact p-values where possible instead of using p < 0.001.

• Avoid subjective words like “highly significant.”

• Provide effect sizes or discuss practical significance, not just statistical significance.

6. Address Temporal Ambiguity (Cross-Sectional Limitations)

Cross-sectional data cannot establish predictors or causality.

Statements such as “ACEs acted as a risk factor” should be softened to:

• “ACEs were associated with…”

• “Participants with prior ACEs had higher odds of distress…”

Replace causal language with associative language consistently.

7. Strengthen Discussion with Theoretical Integration

The discussion should move beyond summarizing results. Please consider:

• Integrating findings with the stress-diathesis model, gender-role theory, or coping and resilience frameworks

• Explaining mechanisms: Why are second/third year students more distressed?

• Discuss sociocultural factors unique to coastal Karnataka that may shape gender differences.

This will bring depth to the paper.

8. Address Potential Measurement Bias

Self-reported ACEs, substance use, and distress are vulnerable to:

• Social desirability bias

• Underreporting

• Recall limitations

Acknowledging and discussing this will improve scientific accuracy.

9. Improve English Language, Structure, and Consistency

Some sentences are long, repetitive, or grammatically inconsistent. Smoother transitions will improve readability.

Examples needing revision:

• “Psychological discomfort” → “psychological distress”

• Repetition in paragraphs discussing substance use changes during COVID-19

• Streamline sentences with clearer academic language

10. Novelty Concern

While the finding that female participants report higher psychological distress is consistent with previous literature, the manuscript should clarify more explicitly what is novel about this study. Since gender differences in mental health have been widely documented globally and in India, the contribution of this study would be strengthened if the authors highlight what unique insights it adds with respect to context, predictors, methodology, or population. I encourage the authors to clearly articulate their study’s novel contribution more explicitly in the introduction and discussion.

Minor Comments

11. Title

The title, in its current form, is slightly long and may appear confusing to readers because it combines several elements (gender differences, prevalence, predictors, location, study design) without a clear logical hierarchy. I recommend simplifying or restructuring the title to enhance clarity and readability. A concise and well-structured title will more accurately capture the study’s focus and improve its appeal to readers.

12. Abstract

• The opening line of the abstract currently states that psychological distress is ‘common’. This wording unintentionally normalizes the problem and does not capture the seriousness of mental health concerns among emerging adults. I recommend rephrasing the first sentence to highlight the public-health significance, rising prevalence, and potential consequences of psychological distress. This will strengthen the scientific tone and underscore the need for the study

• Avoid redundancy between introduction and results.

• Provide exact values for AORs in the abstract if possible.

• Use consistent terminology (“psychological distress,” avoid switching terms).

13. Tables

• Tables are very dense; consider splitting Table 1 for clarity.

• Add brief footnotes explaining abbreviations (AOR, COR).

• Some percentages do not align perfectly with sample numbers—please recheck.

14. Reference Formatting

• Ensure consistent use of APA or journal-required style.

• Several citations have minor errors (capitalization, punctuation, DOI formatting).

• Check for duplicate entries (e.g., Remesan et al. appears multiple times and it also indicates multiple self-citations)

15. Conclusion

Conclusion can be made stronger by:

• Avoiding restating results

• Adding implications for policy (college mental health programs, gender-support interventions)

• Suggesting specific next steps for future research

16. Introduction

“The final paragraph of the introduction states that existing research demonstrates considerable variation in psychological distress globally and within India. However, it is not clear how this claim is connected to the present study’s objectives, as the current research does not appear to analyze or compare global and national prevalence rates. Could the authors clarify whether this statement is directly supported by their data or intended literature scope? If not, I suggest revising the sentence to more accurately reflect the study’s focus.”

17. Predictors in Cross Sectional Research?

“Since the study uses a cross-sectional design, the use of the term ‘predictors may unintentionally imply causal direction, which this design does not support. Would the authors consider using terms such as ‘associated factors’ or ‘correlates’, or alternatively justify their use of the term ‘predictors’ in the context of cross-sectional data?”

---

## [Reviewer Report]

The topic of the manuscript, “Gender Differences in Prevalence and Predictors of Psychological Distress among Emerging Adults in Udupi District, Coastal Karnataka: A Cross-Sectional Study” is interesting. However, it has several shortcomings that should be addressed prior to publication.

-The word “prevalence” appears in both the title and the study aim. However, given that the sample is limited to undergraduate colleges, the authors should reflect this throughout the manuscript (for example, in the title, in the Aim of the Abstract, and in the text on page 5, line 58, and page 6, lines 3 and 4, which reads: “Against this background, the current study is conducted to understand the prevalence and determinants of psychological distress among female and male emerging adults in Udupi district of Karnataka state in India”.

-Subsection “2.2. Measures” should be thoroughly revised to clarify how each variable appearing in the Results section was assessed. Additionally, the Introduction section should explain the rationale for studying each variable presented in the Results section (e.g., religion).

-The descriptions of the World Health Organization Alcohol, Smoking and Substance Involvement Screening Test version three (WHO ASSIST), and the ACE questionnaire should be revised to provide clearer and more complete information on what they assess and how.

- On page 7, lines 31 to 38, the following text referring to the Kessler Psychological Distress Scale (K10) states: “According to this scale, a score below 20 suggests that a participant is likely in a stable psychological state. Scores between 20 and 24 indicate the probable presence of a mild mental disorder; scores between 25 and 29 suggest a moderate mental disorder; and a score of 30 or higher is indicative of a severe mental disorder for the participant”. This text should be revised, as the K10 is not a diagnostic scale but a screening scale, and the correct interpretation of the scores is: 20–24 Likely to have a mild disorder; 25–29 Likely to have a moderate disorder; 30–50 Likely to have a severe disorder.

-Subsection “2.3. Statistical analysis” should be expanded to clearly and comprehensively explain all statistical analyses performed, especially logistic regression.

-Sociodemographic data should be presented in a table, and statistically significant differences between female and male participants should be analyzed.

-Table 1: “Socio-demographic details of the participants in association with psychological distress”, presents data comparing those with “at risk” and “no risk” statuses. It is imperative to explicitly define the criteria for categorizing individuals as “at risk” or “no risk”.

-The Results section should be thoroughly revised, as it is confusing. There also appears to be some errors. For example, on page 12, lines 49 to 60, it states the following: “For female participants, factors such as age, year in which they are studying, type of their educational institution, their living arrangement, whether they believe in religion, their religion, number of siblings, current marital status of parents, substance use, substance use initiation during COVID-19, substance use by family members, peer substance use and ACEs were adjusted in the multivariate analysis after being found significant in the univariate analysis”. However, according to the data in Table 1, several of these variables were not statistically significant in the univariate analyses. For instance, the p-value for “Believe in religion” was 0.109, and the p-value for “number of siblings” was 0.070.

---

## [Reviewer Report]

Comments to the Author:

This manuscript highlights an important issue in the lives of young people, psychological distress. Especially, if there is a notable scarcity of studies examining psychological distress among emerging adults in India. The manuscript is generally clearly written and well structured. I am not qualified to evaluate the language, as I am not a native English speaker, but it appears to be satisfactory. However, I have some comments that you may wish to consider for the next version.

General remarks:

1. The number of decimals in percentages varies, so I suggest using a maximum of one decimal in the text. Even the use of integers can be considered.

2. Word “discomfort” is mentioned three times. Why is it used in these contexts, instead of “distress”, which is used elsewhere throughout the manuscript?

Impact Statement:

1. SDG in mentioned here for the first time. The abbreviation should be explained here or just use the whole name.

Title:

1. The study only concerns college students; this should be made clear in the title.

2. In the title, (also in the abstract, in tables and in the text) it is used term “predicting” psychological stress. However, this is a cross-sectional study (which is mentioned in the title), so we cannot talk about prediction, but rather about the association between things and risk factors. I therefore urge the author to change the wording in this regard.

Abstract:

1. In the Methods, statistical analyses should be mentioned.

Introduction:

1. First reference in the second paragraph (Anyanwu): for clarification, it should be mentioned how many percents of the results were mild and how many were severe symptoms. Also In Ismail’s reference, most cases were mild or moderate. Functional capacity is mainly affected by severe symptoms, while mild symptoms are pretty common, and we are mainly interested in severe symptoms. (from the row 37)

Methods:

1. I would like to know more about college classification, what they are like, especially what “aided” means (page 6, row 31).

2. Has the Kannada version been used elsewhere before? (page 8, row 3)

Results:

1. The sample could also be presented in table form, which would make it easier to compare the figures between variable classes in genders.

2. How is the poverty line defined? (page 8, row 32)

3. Table 1: The meter used (K10) and the score which indicates psychological distress should be indicated in the heading, so that the table can be understood independently without the text. Enter the numbers for men and women in the table, not in the header. For results (p-values) that are not significant, only two decimals is enough.

4. I would like a definition of what mild symptoms and severe symptoms mean. What do they mean in practice in terms of functional capacity? (page 12, row 12) Does mild psychological distress have any practical significance in everyday life? This should be done earlier in the manuscript, in Introduction.

5. Table 2: At first, I didn’t understand this table at all. I think it is perhaps misleading. Can we conclude from this what is the likelihood of mental health problems in the future for women and men students?

6. Table 3: The meter used (K10) and the score which indicates psychological distress should be indicated in the heading, so that the table can be understood independently without the text. The title should not include the word “predictors,” but rather “associations.”

7. The figures shown in the table should not be repeated in the text. (page 14, from row 47)

8. Table 4: The meter used (K10) and the score which indicates psychological distress should be indicated in the heading, so that the table can be understood independently without the text. The title should not include the word “predictors,” but rather “associations.”

Discussion:

1. I would like to focus here only on serious and/or moderate stress. (page 18, row 27)

2. Please pay attention to the length of the paragraphs, as some of them are quite long now.

3. Adverse childhood experiences were the greatest risk for distress in women, so that should be discussed first in Discussion. In men, substance use during COVID was the greatest risk for distress in men and this has been correctly highlighted as the first factor.

4. The gender gap could be addressed in greater depth. What kind of interventions are needed for women and what kind for men?

---

## [Editor Report]

Dear authors,

The reviewers agree that the study is relevant and grounded in substantial fieldwork, but they collectively call for clearer articulation of research aims, theoretical justification for examining gender differences, and tighter alignment between stated objectives and methods. They request clearer sampling rationale and flow, more rigorous and transparent analytic procedures—especially around logistic regression, multicollinearity, confounder selection, and interpretation—and consistent avoidance of causal language given the cross-sectional design. Reviewers also note that several variables in the Results were not statistically significant despite being treated as such, that prevalence claims should be limited to the sampled college population, and that measurement tools need more accurate description, psychometric information, and correct score interpretation. Finally, all reviewers emphasize the need to articulate novelty (given well-established gender patterns), improve English clarity and consistency, strengthen theoretical framing and discussion, ensure accurate terminology (e.g., “associated factors” rather than “predictors”), revise tables for transparency, apply appropriate reference formatting, and more precisely present and interpret findings for publication readiness. Please find the full reviews below.

---

## [Reviewer Report]

Thank you again for inviting me to review the manuscript. I found this manuscript suitable for publication.